# Left Ventricular Assist Device (LVAD)-Related Major Adverse Events Account for a Low Number of Emergency Room Admissions in HeartMate 3™ Patients—A 10-Year Retrospective Study

**DOI:** 10.3390/biomedicines13071702

**Published:** 2025-07-12

**Authors:** Christoph Salewski, Christian Jörg Rustenbach, Spiros Lukas Marinos, Rodrigo Sandoval Boburg, Christian Schlensak, Medhat Radwan

**Affiliations:** Department of Thoracic and Cardiovascular Surgery, University Hospital Tuebingen, Hoppe-Seyler-Strasse 3, 72076 Tuebingen, Germany; christoph.salewski@med.uni-tuebingen.de (C.S.); spiros.marinos@med.uni-tuebinge.de (S.L.M.); rodrigo.sandoval@med.uni-tuebingen.de (R.S.B.); medhat.radwan@med.uni-tuebingen.de (M.R.)

**Keywords:** LVAD, HeartMate3^TM^, emergency room, acute adverse events

## Abstract

**Background:** The yearly number of left ventricular assist device (LVAD) implantations is approximately twice the number of heart transplantations (HTX) in Germany. As the number of patients with an LVAD installed grows, so does the likelihood of their presentation to the emergency room (ER). Due to uneasiness in identifying their primary complaint, ER personnel are often likely to overlook important clues in the treatment of patients with an LVAD. **Methods:** To assess the urgency of patients’ conditions and their relationship with LVADs, we retrospectively examined the ER admissions of patients with HeartMate 3^TM^ (HM 3) LVADs installed between 2014 and 2024 at our university medical center. We counted referrals to the peripheral ward (minor) and to the intensive care unit (ICU, major). Relation to LVAD relation was also recorded. The survival was analyzed with respect to the severity of the cause of admission (minor/major) and the relationship to the LVAD therapy. **Results:** We observed 100 presentations to the emergency department. Of these, 77 were minor and 23 were major. The majority (92) was not related to the LVAD. Of the eight admissions related to the LVAD, two were major adverse events, accounting only for 2% of the total cases. **Conclusions:** An ER presentation of a patient with an HM 3 is very likely to have a medical cause not related to the LVAD. LVAD-related causes were mostly minor and could be treated on the ward.

## 1. Introduction

The yearly number of left ventricular assist device (LVAD) implantations is approximately twice the number of heart transplantations (HTX) in Germany. As the number of patients with an LVAD installed grows, so does the likelihood of their presentation to the emergency room (ER). The Abbott HeartMate 3™ (HM 3) is a third-generation fully magnetically levitated centrifugal LVAD for patients with end-stage heart failure. The first in-human HM 3 implantation was performed in Hannover, Germany in 2014 [1]. Our center of thoracic and cardiovascular surgery performed its first implantation of this device shortly thereafter. For ER personnel, patients with an LVAD form a particular and rare group of patients. ER staff are often uneasy identifying their primary complaint. They focus on the LVAD, and are therefore likely to overlook important clues. To see whether this aversion is justified, we seek to assess the ER presentations of patients with LVADs and identify their underlying causes. Our objective is to determine whether patients with an LVAD constitute a high-risk group when they arrive at the ER. Our investigation is focused on a cohort that uses HM 3.

## 2. Materials and Methods

At the Department for Thoracic and Cardiovascular Surgery of the University Hospital of Tübingen, Germany, we conducted a single-center retrospective study examining the reasons for the presentation to ERs of patients with an HM 3 LVAD and their survival between 2014 and 2024.

### 2.1. Study Design

Over the 10-year course of this retrospective study, 72 patients received an LVAD. Five received the HM II, including one reinstallation. Eleven patients received an HVAD. Overall, 56 patients were implanted with an HM 3 LVAD, non-consecutively. The flow chart for patient inclusion is shown in Figure 1. Eight patients died during the early postoperative period. A total of 48 were discharged and entered the follow-up period.

### 2.2. Metrics

Emergency room admissions were counted during the follow-up period and reported as minor (treated on the ward) or major (referred to the intensive care unit, ICU). From the ER point of view, a minor cause for admission could be transferred to the ward, while a major cause was referred to the ICU. It was recorded whether the presentation was related to LVAD therapy. The clinical courses could differ and deteriorate in the ward or improve on the ICU. The initial referral from the ER was crucial. Baseline parameters (age, sex, implant age, indication, and therapy objective) were recorded. The follow-up time, the time until the first admission to the ER, and survival were calculated.

### 2.3. Statistics

Data are presented as counts, ratios, and percentages in tables and bar charts. Survival analyzes and inferential statistics were performed by means of SPSS 28.0^®^, IBM^®^, and presented as Kaplan–Meier diagrams.

## 3. Results

We present the baseline characteristics, number and causes of admissions to the emergency department, their relation to LVAD therapy, non-presentation-free survival in the emergency department, general survival, and survival with respect to minor and major events.

### 3.1. Baseline Characteristics

A total of 56 patients received an HM 3 LVAD, 45 of whom were male (80.3%). The mean age at implantation was 56.8 +/− 10.9 years and the mean age was 61.2 +/− 11.1 on the day of the last living contact. The mean follow-up time and therefore duration on the LVAD was 4.3 +/− 2.7 years (min, 85 days; max, 9.1 years, range, 8.9 years). Of these, 15 (31.25%) died during the follow-up period. Two patients received heart transplants and spent 2.04 and 4.77 years on HM 3 prior to their transplant. Until 31 December 2024, the 31 (65.6%) ongoing patients spent 4.5 +/− 3.2 years on the device and counting. This does not include the time on other LVADs for the three patients who received an HM 3 as a second LVAD. One patient spent 7.02 years on an HM II until a critical drive line cable defect occurred. Since then, he has been on the HM 3 for 2.2 years and counting. A second patient had an HM II for 2.8 years before suffering a direct device infection. He spent 5.8 more years on an HM 3 until he died of non-LVAD-related sepsis. The third patient spent 3.07 years on an HVAD before experiencing multiple pump thromboses. He has been active on an HM 3 for the last 2.7 years and counting. The last documented patient contact before 31 December 2024 was counted. The study population consisted of 48 (100%) patients, 40 (83%) of whom were male (see Table 1). During the follow-up period, we observed 210 LVAD patient years.

### 3.2. Emergency Room Admissions

In the follow-up period, 37 patients with an HM 3 (77%) presented to the emergency room and 100 admissions occurred. Of these, 77 were minor events that were managed on the ward. Only six of the minor events were related to LVAD therapy. The same cohort suffered 23 major adverse events, but only two were attributed to their LVAD. In other words, of the 100 admissions to the emergency department, only 8 were related to the LVAD and 92 were medical (see Table 2). Of the 100 admissions, 94% could be discharged after treatment. Six patients died after being admitted with major causes that needed referral to the ICU (accounting for 12.5% of patients and 6% of ER admissions). None of the deaths were LVAD-related. However, patients with an LVAD-related reason for presentation to the ER had a higher likelihood of being transferred to the ICU (odds ratio: 1.127).

The leading cause of overall admissions was arrhythmia. The total of 24 arrhythmic events were traced back to only 11 patients, 1 of whom had 9 presentations, another patient had 5, and another had 2 presentations. Infections were the second most common reason for admission. The 15 infections were traced to 10 patients, 2 of whom were admitted 3 times, 1 was admitted 2 times, and 7 were admitted only once. Gastrointestinal complaints were the third most common reason for admission. The 13 GI complaints were suffered by eight patients. Specifically, there were four patients with gall bladder infections, three admissions with diarrhea, one with appendicitis, one with a hernia, one with nausea and vomiting, and three cases of unspecific abdominal pain. In half of the reasons for admission, patients visited the ER multiple times or at least twice for the same reason. The causes of admission are listed in Table 3.

Furthermore, we analyzed where the HM 3 patients were referred and treated. An infection could have been a minor cause and treated conservatively on the ward or a major cause with the need for ICU monitoring and/or surgical treatment. The most common minor causes were arrhythmia (21 cases) and GI bleeding (11 cases). The most common major reasons were infection (six cases) and bleeding (four cases). Driveline infections accounted for most of the minor causes related to LVADs (five cases). In two cases, driveline infections led to systemic inflammation, resulting in sepsis. These were counted as major cases related to LVADs. Table 4 presents the causes of admissions, categorized according to their severity and relation to LVADs.

### 3.3. Survival and Freedom from Emergency Room Admission

In our cohort, 33 (68.8%) of the patients are still alive. The mean survival time was 6.9 years (95% CI: 6.0–7.8 years). The probability of survival is shown in Figure 2a. The probability of living longer than 1568 days (4.3 years) was 80.6%. The probability of living longer than 2355 days (6.5 years) was 51.1%. The longer patients were followed up, the more likely they were to be admitted to the emergency room. Figure 2b shows the estimated time of admission-free survival to the ER as a Kaplan–Meier diagram. The probability of being free from admission to the emergency room for 500 days (1.37 years) is approximately 60%, and probability of remaining free from admission for 1500 days (4.1 years) is slightly above 30%. We then looked at the ER presentations of all 15 deceased patients: when distinguishing between minor and major causes for ER admission, the LVAD patient group with major reasons for admission had a significantly higher probability of dying. This is shown in Figure 2c. The Kaplan–Meier survival estimator was 2047 +/− 153 days for LVAD patients with minor ER admissions, while it was only 1295 +/− 235 days for LVAD patients with major ER admissions. This difference was significant, as the log-rank Mantel–Cox test showed $\left(\chi^{2}=3.869,p=0.049\right)$.

### 3.4. LVAD-RelatedAdmissions to the Emergency Room

#### 3.4.1. Causes of Death

##### General Causes of All Deaths

Of the cohort of 48 patients with an HM 3 LVAD, 15 died during the follow-up period. The most common cause was sepsis (four patients). Three patients had an oncologic complication (one had glioblastoma, one had pancreatic cancer, and one had bladder carcinoma). Two patients had a severe intracranial hemorrhage. A patient developed severe aortic insufficiency and therefore circulatory failure. One patient had pneumonia, while two patients suffered right ventricular failure. One patient developed complicated appendicitis and died of peritonitis. Only a fraction of these patients presented to the ER prior to their death. In some cases, patients had ongoing chronic diseases and decided against further treatment. In other cases, patients were referred from another hospital.

##### Death After Emergency Room Admittance

Five patients died during the clinical course after referral from the ER to the ICU. Therefore, all causes were considered major reasons for admission. None of the causes were related to LVADs. The mean hospital stay from admission to the ER until death was 34.5 days (min, 0 days; max, 84 days). The first patient was admitted with heart failure symptoms as a primary complaint. He was referred to the ICU for his circulatory instability. He had massive aortic regurgitation and died 30 days later. The second patient presented with abdominal pain to the ER. He was diagnosed with appendicitis. However, he suffered peritonitis and died of sepsis. The third patient was admitted with arrhythmia and died 56 days later from right ventricular failure. The fourth patient suffered intracranial hemorrhage and was referred from the ER to the ICU. The bleeding was severe, so the LVAD therapy was discontinued. The fifth patient was referred from the ER to the ICU with pneumonia. Coincidentally, he had severe intracranial bleeding 7 days later, so the LVAD therapy was discontinued.

## 4. Discussion

In our study of 100 admissions to the emergency department of our 48 LVAD patients during the study period, we recorded the causes and severity of the underlying medical condition.

### 4.1. Principal Findings and Clinical Interpretation

This retrospective analysis indicates that LVAD-related presentations to the ER were rare—only 8% of the total admissions of patients with an LVAD—and the major adverse events related to LVADs were exceedingly rare (2%). These figures challenge the assumption that the presence of an LVAD implies a high probability of mechanical failure or an urgent intervention related to the device itself. Most admissions were due to typical health complaints such as arrhythmias, infections, and gastrointestinal disorders, common issues in the cardiac population. This suggests that the presence of an LVAD should not distract clinicians from standard diagnostic routines. Instead, attention should remain focused on the primary complaint.

As could be observed, 92 out of 100 ER presentations of LVAD patients were not related to their LVAD. This should calm ER staff when they are informed of an incoming patient with LVAD. Only 2 of the 23 major cases were related to the LVAD. Patients with major events have a higher likelihood of mortality; however, this is rarely related to the LVAD. ER admissions are equally frequent on weekdays, regardless of the presence of an outpatient LVAD clinic.

Patients with LVAD-related presentations had a slightly higher likelihood of being referred to the ICU (OR 1.127), but LVAD-related causes did not correlate with increased mortality. This highlights that although emergency department personnel may perceive patients with LVADs as high-risk, this assumption should be tempered with clinical judgment and objective findings.

### 4.2. Role of LVAD Clinics and Temporal Presentation Patterns

We investigated whether the availability of an outpatient LVAD clinic influenced ER presentation patterns. We assumed that outpatient clinics for LVAD reduced the number of spontaneous presentations of LVAD patients in the ER department on that day. There was no significant reduction in emergency room visits on clinic days. So, structured outpatient follow-up does not reduce acute presentations for other reasons. However, timing and frequency may not adequately cover periods of vulnerability. However, when there was no LVAD-specialized nurse or perfusionist on duty, such as on weekends, the presentation of LVAD patients in the emergency room was less likely (*p* = 0.008). This may reflect behavioral barriers rather than clinical need. The presentation counts and a binomial analysis are shown in Figure 2d. We encourage emergency room personnel to assess patients with an LVAD in their local standard way, as if there were no LVAD. We advise contacting the local LVAD reference center and informing them of your patient’s admission to the ER if they have an LVAD—if out of hours, at least do this the next day. They will have standard operating procedures in general or patient-specific advice; they can consult over the telephone or take over the patient, if applicable.

### 4.3. Integration with Existing Literature

Patients with LVADs, especially the HeartMate 3, have better survival than those treated with optimal medical therapy [2,3]. Our findings align with the published data from the MOMENTUM3 trial [4], which demonstrated superior survival and fewer adverse events in patients with the HeartMate 3 compared to earlier devices. Similarly, the ELEVATE registry supports the low incidence of device-related complications in the post-market setting [5]. Our study adds evidence that can reassure emergency room personnel: although LVADs remain an important component of heart failure patient therapy, most emergency room presentations are not device-related and can be treated accordingly. Finch et al. examined a 5-year period and found 290 ER presentations of 107 LVAD patients in two adjunct centers in Minnesota and Florida. Overall, 7.9 % were related to the LVAD therapy [6]. Our data comply with these findings, but in a smaller cohort (8% of n = 100 over 10 years). The Heart Failure Association of the European Society of Cardiology curated a position paper for healthcare providers that do not profit from an LVAD center and highlighted bleeding, stroke, and pump thrombosis. They identified bleeding with 0.35–0.65 events per patient year. Neurological events, especially stroke, can occur in 10% of patients during their first year of support, and stroke remains the primary cause of death within the first 6–24 months. Finally, during pump thrombosis, stabilization of the patient is key. Secondary, the location of the thrombus needs to be identified. Medical treatment is available for thrombi in both the inflow and outflow cannula. Surgical options for excision or pump exchange exist [7]. Reza et al. found no significant sex differences in presentation to the ER for device complications, stroke, infection, or heart failure. Women with LVADs presented more frequently with obesity and depression. In addition, respiratory, genitourinary, and gynecological reasons for admission were more frequent. Women form up to 20–25% of the LVAD patient population [8]. In pediatric patients, the percentage of discharges to the outpatient setting post LVAD implantation is much lower than in adults. So, as a non-transplant/non-LVAD center, it is rare to encounter a pediatric LVAD patient in the ER. However, in their study, Pokrajac et al. described that 80% of their 30 discharged patients presented to the ER, with vomiting, abdominal pain, fever, and headache being the most common primary complaints. The mean age of the pediatric patients was 13.5 years, and their study period was 10 years [9]. We advocate for a paradigm shift in emergency care for LVAD patients—from a device-centric to a patient-centric approach. Clinicians should prioritize standard assessment protocols, thoroughly evaluate primary complaints, and then only consider LVAD-specific interventions or consultations. However, centers of reference for LVAD patients would be glad to be informed of the hospitalization of one of their LVAD patients.

### 4.4. Recommendations for Emergency Care

Do not assume an LVAD-related pathology; initial evaluation should follow standard emergency workups.Mental status, vital signs, and primary complaints should guide triage decisions.Contact the LVAD patient reference center when necessary, but avoid unnecessary delays in treatment.Check for anticoagulation, battery status, and LVAD alarms.Standard operating procedures (SOPs) from reference centers can improve decision making (e.g., Royal Brompton Protocol [10] or [11,12], online [13], and for Germany [14,15]).

### 4.5. Limitations and Future Directions

This is a single-center retrospective study, which limits generalization. Also, we categorized admissions as “minor” and “major” based on ER disposition; some clinical courses evolved post-admission and did not reflect initial severity. Our center provides specially trained LVAD nursing staff from 8 a.m. to 4 p.m. on weekdays and a staff of perfusionists on call 24/7. In addition, we operate an outpatient LVAD clinic once a week and see every LVAD patient every three months. All of the mentioned cases were strictly attributed to actual admissions and presentations to the emergency department. We explicitly focused on ER presentations because centers of the basic level of care do not provide an LVAD patient clinic in the same way as we do. Future multi-center prospective studies could explore whether specific risk factors predict ER utilization in this population and whether enhanced patient education or digital monitoring can further reduce unnecessary presentations.

## Figures and Tables

**Figure 1 biomedicines-13-01702-f001:**
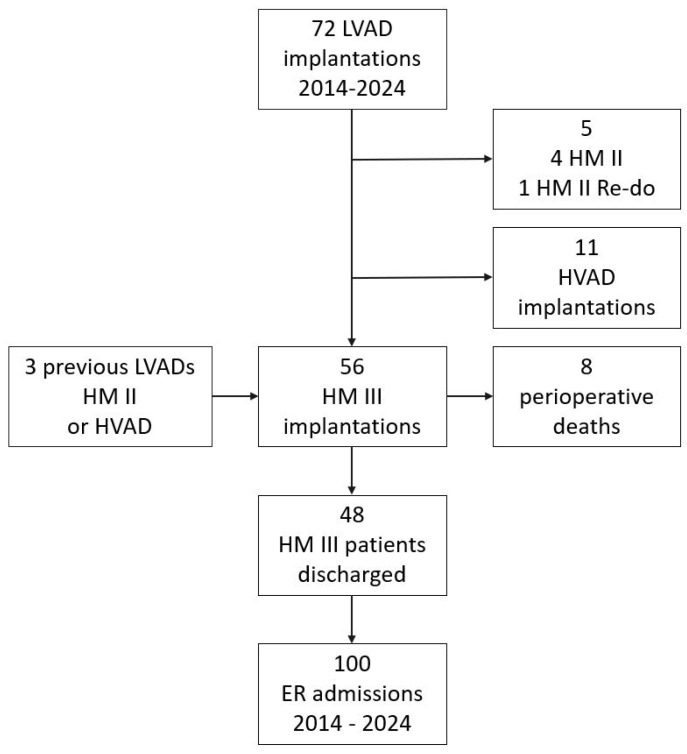
HM 3 implantations between 2014 and 2024 at University Medical Center Tübingen, Germany.

**Figure 2 biomedicines-13-01702-f002:**
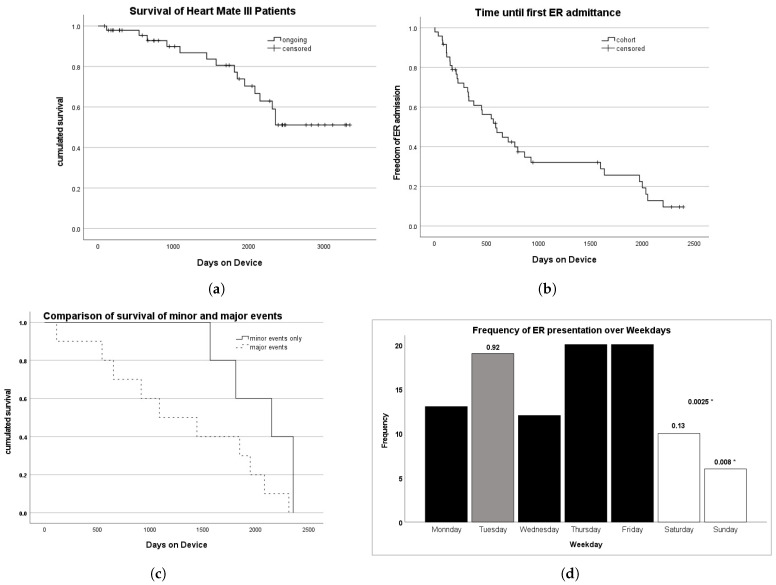
(**a**) Probability of survival with an HM 3 based on our study cohort between 2014 and 2024. The probability of living longer than 1568 days (4.3 years) is 80.6%. The probability of living longer than 2355 days (6.5 years) is 51.1%. (**b**) Probability of ER-admission-free survival with an HM 3 based on our study cohort between 2014 and 2024. Staying free from ER admissions for 500 days had a 60% probability, while staying free from ER admissions for 1500 days had a 30% probability. (**c**) The probability of survival of all 15 deceased patients with respect to minor and major ER admission causes is depicted in a Kaplan–Meier diagram. The dark line represents the group with only minor reasons for ER admission, while the dashed line represents the group with major ER admissions, respectively. LVAD patients with minor reasons for ER admissions had a 100% probability of a 500-day survival, whereas LVAD patients with major reasons had only an 80% probability of survival for the same duration. To live 1500 days with minor ER admission reasons only had a 80% probability, whereas this probability was only 40% for the group with major reasons for ER admission for the same duration. (**d**) Frequency of ER admissions over weekdays. The LVAD outpatient clinic takes place on Tuesdays. On Tuesdays, 19 out of 100 presentations occurred. A binomial distribution analysis showed no fewer presentations in the ER when there was an LVAD outpatient clinic the same day (*p* = 0.92). * Significantly fewer patients presented on Sundays compared to other days (*p* = 0.008). Fewer patients presented on weekends than on weekdays (*p* = 0.0025).

**Table 1 biomedicines-13-01702-t001:** Group demographics and therapy objectives. Counts and % or means +/− SD.

Parameter	n = 48
Gender [m]	40 (83.3%)
Age [y]	61.2 +/− 11.1
Implant age [y]	56.8 +/− 10.9
Follow-up [y]	4.3 +/− 2.7
Indication	
Ischemic cardiomyopathy	24 (50%)
Dilative cardiomyopathy	20 (41.7%)
Myocarditis	4 (8.3%)
Therapy objective	
Bridge to transplant	33 (68.75%)
Destination therapy	11 (22.92%)
Bridge to decision	3 (6.25%)
Bridge to recovery	1 (2.08%)

**Table 2 biomedicines-13-01702-t002:** ER admissions within the follow-up period.

LVAD Relation	Minor	Major	Σ LVAD Relation
No	71	21	92
Yes	6	2	8
Σ severity	77	23	100

**Table 3 biomedicines-13-01702-t003:** Most frequent reasons for ER admission.

Cause	Counts	Patients
Arrhythmia	24	11
Infection	15	10
Gastrointestinal	13	8
Bleeding	7	7
Drive Line Infection	7	6
Neurologic	6	6
Metabolic	5	3
Musculosceletal	5	4
Trauma	5	4
Heart Failure	4	4
Oncologic	2	2
Urologic	2	2
Acute coronary syndrome	1	1
COVID-19	1	1
Pain	1	1
Psychiatric	1	1
Unknown	1	1
Sum	100	72

**Table 4 biomedicines-13-01702-t004:** Causes for ER admission with respect to severity (minor/major) and relation to LVAD.

ER Presentation Causes	Non-LVAD-Related	LVAD-Related	Sum
Minor	71	6	77
Arrhythmia	21		21
Gastrointestinal	10	1	11
Infection	9		9
Drive Line Infection		5	5
Metabolic	5		5
Musculosceletal	5		5
Trauma	5		5
Neurologic	4		4
Bleeding	3		3
Heart Failure	2		2
Oncologic	2		2
Urologic	2		2
COVID-19	1		1
Pain	1		1
Psychiatric	1		1
Major	21	2	23
Infection	6		6
Bleeding	4		4
Arrhythmia	3		3
Drive Line Infection		2	2
Gastrointestinal	2		2
Heart Failure	2		2
Neurologic	2		2
Acute coronary syndrome	1		1
Unknown	1		1
Sum	92	8	100

## Data Availability

The data presented in this study are available on request from the corresponding author.

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
