# Peer review of "Left Ventricular Assist Device (LVAD)-Related Major Adverse Events Account for a Low Number of Emergency Room Admissions in HeartMate 3™ Patients—A 10-Year Retrospective Study"

_biomedicines, 2025, doi:10.3390/biomedicines13071702_

Round 1
Reviewer 1 Report
Comments and Suggestions for Authors
A well written paper. Concise and factual. It is of interest to me that you have so few LVAD insertions in 10 years. It would have been good to include other institutions in your analysis. Maybe that could be the next paper.
Author Response
Comment 1. A well written paper. Concise and factual. It is of interest to me that you have so few LVAD insertions in 10 years. It would have been good to include other institutions in your analysis. Maybe that could be the next paper.
Response 1. Thank you very much for the benevolent assessment of our paper. Indeed we only had 56 implantations in 10 years. We are located in a state with approx 15 mio inhabitants. This area is covered by 5 cardiac surgeries, three of which have an LVAD program and two of which have a transplant program. Therefore we cover about 5 mio inhabitants. Beeing so close to two larger centers with transplant programs causes the little number of yearly implantations. However, our outpatient clinics takes care of LVAD 50 patients that we see on the regular basis described in our paper. Thank you for your suggestion to include other institutions in our analysis. Indeed this could generate data for a next study.
Reviewer 2 Report
Comments and Suggestions for Authors
Thank you for the opportunity to review the manuscript "LVAD-related major adverse events account for a low number of emergency room admissions in HeartMate 3™ patients. A 10-year retrospective study."
Manuscript is very interesting and include balanced and critical view of the research area. This category of patients is one of the most difficult in cardiology practice. And certainly we need more researches in this field.
The authors described the structure of complications, reasons for hospitalization and ER admissions clear.
However, there are several questions:
1. line 75 - "follow-up period, we observed 210 LVAD patient years" - what did autors mean? Is there an error here?
2. line 107 - "In our cohort, 33 (68.8%) of the patients are still alive." - are the data correct?
Since there were initially 56 device implantations. However, if we take into account 56 patients, the survival rate will look worse.
It is clear, authors mean patients after discharge and long-term follow-up after implantation - perhaps this point should be clarified.
3. Was the adherence and quality of drug therapy in these patients assessed after discharge and did these factors affect the development of complications?
4. In my opinion, the reference list is not up-to-date enough. Most of the references are older than 5 years. There are a number significant results and registers in both Europe and USA which demonstrate positive experience in LVAD-patients.
Author Response
Comment 1. line 75 - "follow-up period, we observed 210 LVAD patient years" - what did autors mean? Is there an error here?
Respone 1: Thank you for your comment. Our study covers a 10 year period. Within tis period, one patient lived 9.16 years with an LVAD, another patient lived 9.08 years, a third patient lived 8.98 years... the last patient included was implanted with an LVAD last year, so he adds only 0.23 years to the study. In total, out 48 patients add 210 years of LVAD follow-up to this study. The unit of the follow up duration is "patient years". Patient-years (or person-years) is a unit of measurement used in epidemiology and clinical research to represent the total time a group of patients is observed or treated in a study, taking into account the varying lengths of time individuals participate. It's essentially the sum of the time each patient contributes to the study, regardless of whether they were in the study for the entire duration or only part of it.
Comment 2. line 107 - "In our cohort, 33 (68.8%) of the patients are still alive." - are the data correct?
Since there were initially 56 device implantations. However, if we take into account 56 patients, the survival rate will look worse. It is clear, authors mean patients after discharge and long-term follow-up after implantation - perhaps this point should be clarified.
Response 2. Pokajac et. al reported ER presentations of pediatric LVAD patients in a 10-year period. They counted 104 implantations, but only 30 (28.8%) survived for outpatient care. Of those 30 patients, 24 (80%) account for the population that presented to the OR. So the style of reporting is the same and appropriate. But you are right. Initially we implanted 56 patients, many of them in critical INTERMACS levels. 8 did not survive the postoperative phase and were never discharged from intensive care. Taking into account the total number of implantations 33 surviving patints account for only 59% of all 56 implanted patients. However, this study focuses on emergency room presentations of LVAD patients from the outpatient setting. The 8 deceased (14%) of the initial 56 implanted patients were never discharged and could therefore never present to an emergency room. We do not try to make our data look good. We just want to describe the ratio of patients that survived until today of all the patients that were discharged. This is 33/48 (68.75%).
Comment 3. Was the adherence and quality of drug therapy in these patients assessed after discharge and did these factors affect the development of complications?
Response 3: Important question. The large majority of ER presentations account for patients that were implanted at our University Medical Center. All of these patients present on a 3-month basis to our LVAD outpatient clinic. Their protocol is blood works, EKG, LVAD Echo, Download of LVAD history from the controller, and a presentation to a physician (cardiac surgeon and/or cardiologists). So the medication is regularly updated for adherance, side effects, and patient conditions. However, you are right assume a number of ER presentations is due to incompliance to oral medication. Unter this protocol compliance to medication is well, in our experience.
Comment 4. In my opinion, the reference list is not up-to-date enough. Most of the references are older than 5 years. There are a number significant results and registers in both Europe and USA which demonstrate positive experience in LVAD-patients.
Response 4: Your comment enhances our manuscript. Thank you. We did an additional pubmed search with the terms "emergency room presentation of LVAD patients" and limited the publication to the years between 2020 and 2025. We identified 9 results. Four of which were relevant to our manuscript. They span one epidemiological paper like ours, one position paper of the European Society of Cardiologists for emergency room management, one focusing on gender differences, and a last one focusing on pediatric patients. We added them to our discussion.